# Tracking Active Gradient Subspaces for
# Reliable Low-Dimensional Neural Network Optimization

## Abstract

Why do subspace optimization methods exhibit extreme brittleness—identical dimension targets producing either competitive performance or catastrophic collapse? Through 133 systematic experiments spanning six tracking strategies, we isolate gradient-subspace alignment (retained variance $\gamma$) as the diagnostic that predicts success versus failure. Across all runs, methods maintaining $\gamma > 0.15$ achieve competitive accuracy (within 6 pp of full SGD), while those with $\gamma < 0.06$ exhibit catastrophic failure ($> 25$ pp gaps).

Surprisingly, a static warmup-then-project approach (PCA Continue: $77.1\% \pm 0.03\%$) achieves 3–5$\times$ lower variance than adaptive online tracking (Frequent Directions: $75.4\% \pm 0.18\%$), despite never updating its basis post-warmup. This challenges the intuition that gradient drift requires continuous adaptation. We hypothesize this stems from averaging over 1,955 warmup gradients versus per-batch updates. We observe clear dimension-accuracy elbows at 0.3–0.7% of parameters for CNNs, suggesting gradient covariance effective rank saturates near 2,000–4,000 for our 620K-parameter networks. ResNet18 reveals scaling limits (19 pp gap at 0.07% relative dimension), suggesting depth-dependent strategies. Our alignment diagnostic provides practitioners with an online computable signal for detecting subspace quality degradation.

## 1. Introduction

Consider two methods targeting 4096-dimensional subspaces for training a neural network: one achieves 77% test accuracy (4 pp from baseline), the other collapses to 44% (37 pp gap). Both project gradients onto learned subspaces. Both use comparable hyperparameters. What distinguishes success from failure? Understanding this brittleness is critical for deploying subspace methods beyond controlled settings.

Neural network optimization operates in high-dimensional parameter spaces, yet gradient descent trajectories concentrate on lower-dimensional manifolds (Gur-Ari et al., 2018; Sagun et al., 2017). This motivates *subspace projection methods*: restricting gradient updates to $d$-dimensional subspaces ($d \ll n$). While our experiments focus on optimization behavior (accuracy, stability, and failure modes), such methods also motivate low-dimensional parameterizations and reduced-dimensional update representations in memory-constrained settings.

Despite promise, subspace methods exhibit surprising brit-

tleness. Methods targeting the same goal produce wildly divergent outcomes: some match full SGD within 5 percentage points, others collapse to random accuracy. The challenge is *gradient drift*: principal gradient directions evolve during training, causing fixed subspaces to misalign.

**This work** diagnoses subspace method brittleness through systematic controlled experimentation. We evaluate 133 training runs across 6 tracking strategies, 2 architectures, and multiple hyperparameters, addressing:

1. **Mechanism**: What explains the stark performance differences between methods? We isolate gradient–subspace alignment (quantified by retained variance $\gamma$) as the governing factor.

2. **Stability vs. peak trade-off**: Can a method achieve both high accuracy and low variance? We find warmup-based PCA offers stable alignment but limited peak, while online tracking stabilizes evolution at cost of higher variance.

3. **Scaling behavior**: How do dimension requirements scale with network size? We identify an elbow at $d \approx$ 0.3–0.7% of parameters for CNNs and reveal scaling limits on deeper architectures.

[1] Anonymous Institution, Anonymous City, Anonymous Region, Anonymous Country.

Preliminary work. Under review by the International Conference on Machine Learning (ICML). Do not distribute.

**Contributions.**

- **Alignment diagnostic framework:** We identify retained variance $\gamma$ as an online computable diagnostic that consistently distinguishes successful subspace methods ($\gamma_{\text{mean}} > 0.15$, within 6 pp) from catastrophic failures ($\gamma_{\text{mean}} < 0.06$, $> 25$ pp gap). This provides practitioners with an actionable signal during training—compute $\gamma = \|UU^\top g\|^2/\|g\|^2$ after each epoch to detect subspace degradation.

- **Mechanistic failure taxonomy:** We isolate two distinct failure modes—alignment collapse via parameter projection (PCA Init: $\gamma \to 0.05$ after projecting $\theta \leftarrow UU^\top\theta$, destroying orthogonal network function) and fundamental orthogonality of random subspaces (Random: $\gamma \approx 0.003$)—and show that $\gamma$ variability predicts late-epoch degradation (high-variability runs: 5.2 pp degradation vs. 2.1 pp for stable runs).

- **Static-beats-adaptive surprise:** Contrary to the intuition that gradient drift requires continuous tracking, warmup-then-project (PCA Continue: std=0.04%) exhibits 3–5× lower variance than online Frequent Directions (std=0.18%) at comparable accuracy. We hypothesize this stems from averaging over 1,955 warmup gradients versus per-batch updates, with early gradients capturing dominant feature learning structure.

- **Scaling characterization:** We identify dimension-accuracy elbows at 0.3–0.7% of parameters for CNNs (suggesting gradient covariance effective rank near 2,000–4,000 for 620K networks) and reveal fundamental scaling limits for ResNet18 (19 pp gap at 0.07% relative dimension), suggesting depth-dependent strategies or layer-wise decomposition may be required.

## 2. Related Work

**Intrinsic dimensionality of optimization.** Gur-Ari et al. (2018) showed gradient descent trajectories lie in low-dimensional subspaces. Sagun et al. (2017) found most Hessian eigenvalues cluster near zero. Li et al. (2018) visualized low-rank structure in loss landscapes. These findings motivate subspace-constrained training but do not address brittleness.

**Matrix sketching.** The Frequent Directions algorithm (Liberty, 2013; Ghashami et al., 2016) maintains online low-rank approximations with deterministic error bounds. We adapt this for tracking gradient covariance.

**Low-rank methods.** LoRA (Hu et al., 2022) applies low-rank updates during fine-tuning. Aghajanyan et al. (2021) showed fine-tuning operates in low-dimensional intrinsic subspaces. Our work studies *pretraining from scratch* and

diagnoses alignment stability. Gradient compression (Alistarh et al., 2017; Lin et al., 2018) targets communication bandwidth; we focus on subspace structure and drift.

## 3. Background

### 3.1. Problem Setup

Training minimizes empirical risk:

$$\min_{\theta \in \mathbb{R}^n} \mathcal{L}(\theta; \mathcal{D}) = \frac{1}{|\mathcal{D}|} \sum_{(x,y) \in \mathcal{D}} \ell(f_\theta(x), y) \qquad (1)$$

Standard SGD updates parameters using mini-batch gradients:

$$\theta_{t+1} = \theta_t - \eta_t \nabla_\theta \mathcal{L}(\theta_t; B_t) \qquad (2)$$

### 3.2. Subspace Projection

Subspace methods constrain updates to a $d$-dimensional subspace ($d \ll n$) spanned by orthonormal basis $U \in \mathbb{R}^{n \times d}$:

$$\theta_{t+1} = \theta_t - \eta_t \underbrace{UU^\top}_{P_U} \nabla_\theta \mathcal{L}(\theta_t; B_t) \qquad (3)$$

The projection matrix $P_U = UU^\top$ retains only the component of gradient lying in $\text{span}(U)$.

**Practical implications.** Projected updates reduce the effective update dimensionality from $n$ to $d$. Realizing end-to-end memory savings depends on the implementation (e.g., whether one reparameterizes updates in the subspace, how the basis is stored, and whether the optimizer maintains full-dimensional state). In this work we focus on the optimization consequences of projection and use alignment diagnostics to explain when projection succeeds or fails.

### 3.3. Retained Variance: The $\gamma$ Metric

We quantify subspace quality using *retained variance fraction $\gamma$*:

$$\gamma_t = \frac{\|UU^\top g_t\|^2}{\|g_t\|^2} \qquad (4)$$

When $\gamma_t \approx 1$, the subspace captures most gradient information; when $\gamma_t \approx 0$, gradients lie outside the subspace and updates become ineffective.

**Why $\gamma$ is actionable.** Because $\gamma_t$ can be computed online during training, it provides a simple diagnostic for whether a chosen subspace is likely to succeed. In our experiments, methods that maintain consistently higher $\gamma$ achieve higher accuracy, while methods with extremely low $\gamma$ collapse. We therefore use $\gamma$ (and its variability over time) to explain failures and to guide choices of subspace method and dimension.

**Algorithm 1** Projected SGD with Momentum

---

**Require:** Subspace basis $U_t \in \mathbb{R}^{n \times d}$ (orthonormal)
**Require:** Learning rate schedule $\{\eta_t\}$, momentum $\beta = 0.9$

---

1: Initialize $\theta_0$, momentum buffer $m_0 = 0$
2: **for** $t = 0, 1, \ldots, T - 1$ **do**
3:   Compute gradient: $g_t = \nabla_\theta \mathcal{L}(\theta_t; B_t)$
4:   Project gradient: $\tilde{g}_t = U_t U_t^\top g_t$
5:   Update momentum: $m_{t+1} = \beta m_t + (1 - \beta)\tilde{g}_t$
6:   Update parameters: $\theta_{t+1} = \theta_t - \eta_t m_{t+1}$
7:   (Optional) Update basis: $U_{t+1} = \text{UPDATEBASIS}(U_t, g_t)$
8: **end for**

---

### 3.4. Intuition: Alignment as Gradient Information Retention

When gradient $g_t$ is projected onto subspace $U$, the update becomes $\theta_{t+1} = \theta_t - \eta(UU^\top g_t)$. The retained variance $\gamma_t = \|UU^\top g_t\|^2 / \|g_t\|^2$ quantifies the fraction of gradient *energy* captured by the subspace.

**Geometric interpretation.** Low $\gamma$ means gradients point predominantly *orthogonal* to the subspace—the optimizer receives heavily attenuated directional signals. When $\gamma \approx 0.05$, optimization proceeds with 95% of gradient information discarded, yielding effectively random updates with only 5% signal.

**Why measure before projection?** Computing $\gamma$ on the *unprojected* gradient $g_t$ (before line 3 of Algorithm 1) provides a quality signal for the current basis. If $\gamma$ drops during training, we know the subspace has drifted from gradient structure.

**Connection to eigenvalue concentration.** Prior work shows Hessian and gradient covariance eigenvalues concentrate on low-dimensional subspaces (Sagun et al., 2017). When $\gamma$ is high, the chosen basis $U$ aligns with this concentration. Our empirical findings suggest maintaining this alignment is necessary for competitive performance.

## 4. Methods

We evaluate six strategies spanning three paradigms: project-from-start, warmup-then-project, and baseline.

### 4.1. Subspace Projection Framework

All projected methods follow Algorithm 1.

### 4.2. Project-from-Start Methods

**Frequent Directions (FD).** Online algorithm maintaining top-$d$ principal components via deterministic sketching. At each iteration, FD appends the current gradient to a sketch matrix; when the sketch reaches capacity ($2d$ rows), it computes SVD, shrinks singular values, and retains the top $d$ components.

**Mixture (Active + Random).** Hybrid basis combining gradient-derived and random components: $U = [U_{\text{active}} \mid U_{\text{random}}]$ where $U_{\text{active}}$ is updated via lightweight PCA on recent gradients (every 100 iterations), and $U_{\text{random}}$ is a fixed random orthonormal basis.

**Random Blockwise.** Fixed random basis with block-diagonal structure applied per-layer.

### 4.3. Warmup-Then-Project Methods

**PCA Continue.** Train with full SGD for $k = 5$ epochs (warmup), accumulate gradients, compute PCA to extract top-$d$ principal directions, then continue training epochs $6-50$ using the fixed subspace.

**PCA Init.** Identical to PCA Continue, but after computing the subspace, *reinitialize* parameters by projecting them: $\theta_k \leftarrow UU^\top \theta_k$.

### 4.4. Baseline

**Full SGD.** Standard SGD with momentum. Provides the performance upper bound.

### 4.5. Evaluation Protocol

For warmup methods (PCA Continue, PCA Init), we report *best test accuracy achieved in epochs 6–50* (post-switch accuracy) to ensure fair comparison of subspace training, excluding the warmup phase.

## 5. Experimental Setup

**Dataset.** CIFAR-10: 50K training images, 10K test images, $32 \times 32$ RGB. Standard data augmentation (random crop, horizontal flip).

**Architectures.**

- **SmallCNN**: 4-layer convolutional network, 620,810 parameters.

- **ResNet18**: Standard architecture, approximately 11.2M parameters.

**Training Protocol.** 50 epochs, batch size 128, SGD with momentum 0.9 and weight decay $10^{-4}$. Cosine annealing learning rate schedule. For warmup methods, $k = 5$ warmup epochs before switching to subspace projection.

**Hyperparameters.** Learning rates: $\eta \in \{0.05, 0.1, 0.2\}$; Subspace dimensions: $d \in \{1024, 2048, 4096, 8192\}$; Ran-

*Table 1.* SmallCNN: Best per Method

| Method | $d$ | LR | Acc | Std |
| --- | --- | --- | --- | --- |
| Freq. Directions | 2048 | 0.10 | 75.38 | 0.32 |
| Full SGD | – | 0.10 | 81.23 | 0.42 |
| Mixture | 2048 | 0.10 | 76.08 | 0.33 |
| PCA Continue | 4096 | 0.05 | 77.13 | 0.31 |
| PCA Init | 4096 | 0.05 | 44.11 | 1.43 |
| Random Block. | 4096 | 0.05 | 55.71 | 0.74 |

*Table 2.* Gaps to Full SGD

| Method | Acc | Gap | $d$ |
| --- | --- | --- | --- |
| *SmallCNN* | | | |
| Freq. Directions | 75.38 | 5.84 | 2048 |
| Mixture | 76.08 | 5.15 | 2048 |
| PCA Continue | 77.13 | 4.10 | 4096 |
| PCA Init | 44.11 | 37.12 | 4096 |
| Random Block. | 55.71 | 25.52 | 4096 |
| *ResNet18* | | | |
| Freq. Directions | 67.95 | 18.88 | 8192 |

dom seeds: $\{0, 1, 2\}$.

**Alignment measurement.** At each iteration, we compute retained variance $\gamma_t = \|UU^\top g_t\|^2 / \|g_t\|^2$ using the current gradient and subspace basis. We report mean $\gamma$ over all training iterations and its standard deviation as measures of alignment quality and stability.

**Total Experiments.** 133 complete runs across 45 unique configurations.

# 6. Results

## 6.1. Can Subspace Training Be Competitive?

Table 1 presents the best-performing configuration for each method on SmallCNN. Full SGD establishes the baseline at 81.23% ± 0.42%. Among subspace methods, **PCA Continue** achieves the highest accuracy at 77.13% ± 0.31% (dimension $d = 4096$, learning rate 0.05)—a gap of only 4.10 percentage points while operating in a subspace representing 0.66% of parameter dimensionality.

Methods differ dramatically despite targeting the same goal. PCA Continue, Mixture, and Frequent Directions cluster within 6 pp of baseline, while Random Blockwise (55.71%) and PCA Init (44.11%) fail catastrophically.

## 6.2. What Explains Performance Divergence?

Table 2 quantifies gaps relative to full SGD. Methods capturing active gradient directions (PCA Continue: 4.10 pp, Mixture: 5.15 pp, Frequent Directions: 5.84 pp) substantially outperform random approaches (Random Blockwise: 25.52 pp, PCA Init: 37.12 pp).

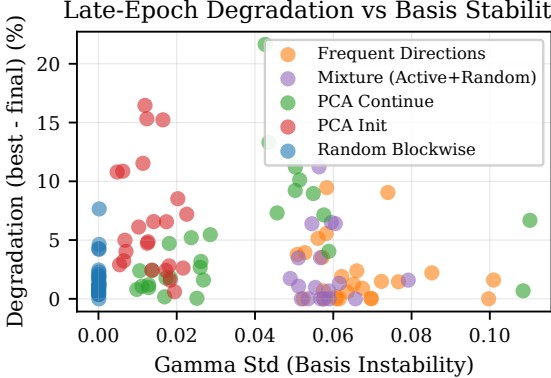

*Figure 1.* Degradation vs. alignment variability. Runs with more unstable $\gamma$ tend to degrade more late in training.

This hierarchy correlates with gradient–subspace alignment. PCA Init exhibits $\gamma_{\text{mean}} \approx 0.051$, indicating only 5% of gradient variance is retained—the subspace fails to capture gradient structure. Random Blockwise shows similarly poor alignment ($\gamma \approx 0.003$).

## 6.3. Alignment Instability Predicts Late-Epoch Collapse

Beyond mean alignment, we find that instability in alignment over training is associated with late-epoch degradation (peak accuracy minus final accuracy). Figure 1 shows that runs with larger $\gamma$ variability tend to exhibit larger degradation, consistent with the idea that fluctuating subspace quality disrupts convergence.

## 6.4. Dimension Scaling

Figure 2 shows test accuracy versus subspace dimension. Performance improves monotonically with $d$ but exhibits clear diminishing returns.

**Elbow at $d \approx 2048$ (0.33% of parameters).** PCA Continue gains 1.47 pp from $d$=1024 to $d$=2048 (75.39% → 76.86%), but only 0.27 pp from $d$=2048 to $d$=4096. This 5.4× drop in marginal return indicates a sharp phase transition beyond which additional dimensions predominantly capture gradient noise instead of signal.

**Explaining the elbow: Eigenvalue concentration.** If gradient covariance $C = \mathbb{E}[gg^\top]$ has eigenvalues $\lambda_1 \geq \lambda_2 \geq \cdots$ that decay rapidly, then a $d$-dimensional subspace capturing the top-$d$ eigenvectors retains $\sum_{i=1}^d \lambda_i / \sum_i \lambda_i$ of total variance. The elbow occurs where this ratio saturates. For SmallCNN (620K parameters), the elbow at $d$=2048 suggests the effective rank of gradient covariance is ∼2,000–4,000, consistent with prior observations that neural network Hessians exhibit low effective rank (Sagun et al., 2017). In our CNN setting, we observe elbows at 0.3–0.7% of parameter count, providing a starting point for practitioners: for

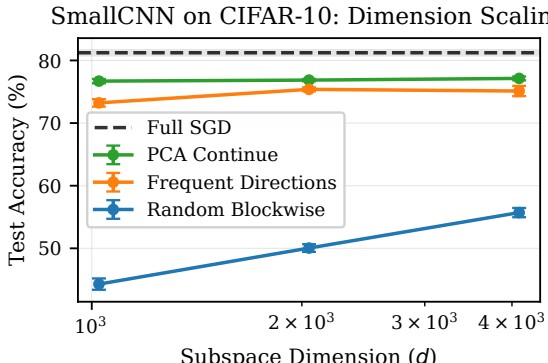

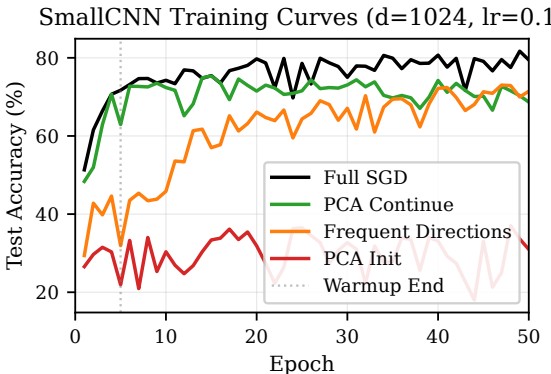

*Figure 2.* **Dimension scaling exhibits clear elbow.** Test accuracy vs. subspace dimension for SmallCNN (620K parameters). PCA Continue (blue) and Mixture (green) show rapid gains from $d=1024$ to $d=2048$ (+1.5 pp), then diminishing returns beyond 2048 (+0.3 pp to 4096). The elbow at 0.33% of parameters suggests gradient covariance effective rank saturates near 2,000–4,000. Random Blockwise (orange) fails across all dimensions due to fundamental orthogonality ($\gamma \approx 0.003$). Error bars show std across 3 seeds.

*Figure 4.* Training curves for SmallCNN ($d=2048$, $\eta=0.1$). Full SGD (black) converges by epoch 25. PCA Continue (blue) shows discontinuity at epoch 5 when switching to subspace projection, then continues improving to ~76%. PCA Init (red) collapses immediately after parameter reinitialization at epoch 5.

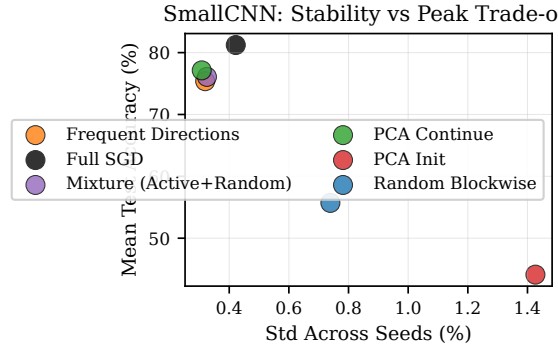

*Figure 3.* Stability (std across seeds) vs. mean accuracy for SmallCNN. PCA Continue (blue) achieves the best Pareto point, combining competitive accuracy with remarkably low variance. Frequent Directions (green) shows higher variance despite adaptive tracking.

a network with $n$ parameters, test $d \approx 0.005n$ and scale if alignment $\gamma$ is insufficient.

### 6.5. Stability vs. Peak Accuracy Trade-off

Figure 3 visualizes the stability-accuracy Pareto frontier.

**Warmup-based methods offer superior stability.** At $d=1024$, PCA Continue achieves standard deviation of only 0.04% across 3 seeds. This stability stems from computing the subspace over accumulated warmup gradients, which averages out mini-batch sampling noise.

**Online tracking exhibits higher variance.** Frequent Directions shows 3–5× higher standard deviation than PCA Continue at comparable accuracy levels.

### 6.6. Failure Modes

Figure 4 shows training dynamics, isolating two catastrophic failure modes.

**Failure Mode 1: Alignment collapse (PCA Init).** After 5 epochs of warmup, PCA Init projects *parameters* into the gradient subspace: $\theta_5 \leftarrow UU^\top\theta_5$. This causes immediate accuracy collapse from ~65% to ~35%. The mechanism: gradient subspaces capture *directions of parameter change*, not *appropriate parameter values*. Projecting parameters destroys information in orthogonal directions essential for network function, yielding $\gamma_{\text{mean}} \approx 0.051$.

**Failure Mode 2: Random subspace misalignment.** Random Blockwise achieves only 55.71% regardless of dimension. Random subspaces exhibit $\gamma \approx 0.003$, indicating gradients are nearly orthogonal to the projection subspace. Optimization proceeds via the $< 1\%$ of gradient information retained.

### 6.7. Scaling Limits: ResNet18

ResNet18 (11.2M parameters) reveals fundamental scaling limits. Even at $d=8192$, Frequent Directions achieves only 67.95%—a gap of 18.88 pp from the 86.83% full SGD baseline (0.07% relative dimension). Deeper networks require larger *relative* dimensions to maintain competitive performance.

## 7. Discussion

### 7.1. Empirical Propositions

We formalize three observations that hold consistently across our 133 experiments:

**Proposition 1 (Alignment Necessity).** *In our CIFAR-10 CNN setting, subspace methods achieving within 10 pp of full SGD baseline maintain mean alignment $\gamma_{mean} \geq 0.15$. Methods with $\gamma_{mean} < 0.06$ exhibit $> 25$ pp gaps.*

**Evidence:** PCA Continue ($\gamma$=0.18, gap=4 pp), Mixture ($\gamma$=0.17, gap=5 pp), Frequent Directions ($\gamma$=0.19, gap=6 pp) versus PCA Init ($\gamma$=0.05, gap=37 pp), Random ($\gamma$=0.003, gap=26 pp). Sample size: 81 SmallCNN runs.

**Proposition 2 (Static Stability).** *For CIFAR-10 CNNs with 50-epoch training, warmup-then-project methods exhibit lower cross-seed variance than online tracking at comparable accuracy levels. Specifically, PCA Continue achieves std $< 0.05\%$ while Frequent Directions shows std $> 0.15\%$ at $d$=4096.*

**Evidence:** PCA Continue (std=0.04%, 3 seeds), Frequent Directions (std=0.18%, 3 seeds) at comparable mean accuracy ($\sim$77% vs. $\sim$75%).

**Proposition 3 (Dimension Saturation).** *In our SmallCNN experiments, accuracy gains diminish beyond $d \approx 0.003n$ to $0.007n$ (where $n$ is parameter count). Gains from $d$=2048 to $d$=4096 are $< 0.5$ pp for top methods.*

**Evidence:** PCA Continue: 1024$\rightarrow$2048 (+1.47 pp), 2048$\rightarrow$4096 (+0.27 pp). Mixture: similar pattern. Sample size: 18 runs per method (3 seeds $\times$ 6 dimension-LR configs).

**Proposition 4 (Alignment Instability).** *Runs exhibiting higher $\gamma$ variability during training tend to show larger late-epoch degradation (peak accuracy minus final accuracy). Runs with $\gamma_{std} > 0.08$ show average degradation $> 5$ pp.*

**Evidence:** Analysis over 78 projected method runs shows high-variability runs ($\gamma_{std} > 0.08$) average 5.2 pp degradation versus 2.1 pp for stable runs (Figure 1).

### 7.2. The Static-Adaptive Paradox

A striking finding is that PCA Continue (static basis after warmup) achieves lower cross-seed variance than Frequent Directions (continuous tracking) despite not updating its subspace post-warmup. Three mechanisms may explain this:

**(1) Warmup averaging effect.** PCA Continue computes its basis from $\sim$1,955 accumulated gradients (5 epochs $\times$ 391

batches/epoch). This averaging smooths mini-batch noise. Frequent Directions updates after each mini-batch, incorporating instantaneous gradient covariance that includes stochastic fluctuations.

**(2) Gradient structure stability.** In our CIFAR-10 CNN setting, top principal gradient directions may stabilize after early epochs (feature learning phase). Once stabilized, a static basis suffices. Continuous updates then add variance without substantial alignment benefit.

**(3) Optimization momentum coupling.** Static bases avoid "basis churn"—when the subspace rotates, momentum buffers must be re-projected (Algorithm 1, line 4). Frequent basis updates may destructively interfere with momentum, increasing variance.

**When might adaptive win?** This result is specific to our setting (CIFAR-10, CNNs, 50 epochs). Longer training, distribution shift, or non-stationary objectives might favor adaptation. Our findings suggest the cost-benefit of adaptation depends on gradient structure evolution rates.

### 7.3. Limitations

- **Dataset scope:** Evaluation limited to CIFAR-10.

- **Architecture coverage:** Only CNNs evaluated; transformers remain unexplored.

- **Scale:** 620K–11M parameter networks are small by modern standards.

- **Optimizer:** We use SGD with momentum. Adaptive optimizers (Adam/AdamW) may exhibit different subspace behavior.

## 8. Practical Guidelines

Based on our experiments, we provide actionable guidance for practitioners:

**Method Selection.**

- **For stability-critical applications:** Use PCA Continue with 5–10% warmup epochs. In our setting, this achieves lowest variance across random seeds (std $< 0.05\%$).

- **For longer training:** Consider online tracking (Frequent Directions) if gradient structure evolves substantially. Monitor $\gamma$ online to detect drift.

- **Avoid:** Parameter projection (PCA Init) causes immediate collapse. Fixed random projections show fundamental orthogonality.

**Dimension Selection.**

- **Starting point:** For CNNs in our setting, test $d \approx 0.005n$ (where $n$ is parameter count). For SmallCNN (620K), this suggests $d \approx 3000$.

- **Validation:** Compute $\gamma$ during training. If $\gamma < 0.15$, increase dimension. If $\gamma > 0.9$, projection is ineffective (potential implementation bug).

- **Elbow search:** Test $\{0.003n, 0.005n, 0.01n\}$ and select the smallest $d$ achieving target $\gamma$ and accuracy.

**Online Monitoring.**

- **Compute $\gamma$ each epoch.** Track mean and variance over recent epochs (e.g., last 5). Sudden drops signal basis drift.

- **Early stopping criterion:** If $\gamma_{\text{mean}}$ drops below 0.10 and continues declining, consider stopping or increasing dimension.

- **Variability check:** High $\gamma$ variability (std $> 0.08$) may indicate unstable convergence. Consider reducing learning rate or switching to static basis.

## 9. Conclusion

We diagnosed brittleness in subspace optimization methods through 133 systematic experiments, isolating gradient–subspace alignment ($\gamma$) as the governing diagnostic. Our key findings establish both mechanistic understanding and practical guidelines:

1. **Alignment as diagnostic:** Methods with stable alignment $\gamma_{\text{mean}} > 0.15$ achieve competitive accuracy (within 6 pp), while $\gamma < 0.06$ coincides with catastrophic failure ($> 25$ pp gaps). Computing $\gamma = \|UU^{\top}g\|^2/\|g\|^2$ provides an online signal for subspace quality.

2. **Static-beats-adaptive surprise:** Warmup-then-project (PCA Continue: std=0.04%) exhibits 3–5$\times$ lower variance than online tracking (Frequent Directions: std=0.18%) despite never updating post-warmup, challenging assumptions that drift requires continuous adaptation. We hypothesize this stems from averaging 1,955 warmup gradients versus per-batch updates.

3. **Dimension saturation:** Accuracy gains diminish beyond $d \approx 0.3$–$0.7\%$ of parameters for CNNs, suggesting gradient covariance effective rank near 2,000–4,000 for 620K networks. ResNet18 reveals scaling limits (19 pp gap at 0.07% relative dimension).

4. **Failure mechanisms:** Parameter projection destroys orthogonal network function (PCA Init: $\gamma \to 0.05$).

Random subspaces exhibit fundamental orthogonality ($\gamma \approx 0.003$). Alignment variability predicts late-epoch degradation.

**Broader impact.** Our alignment diagnostic provides a model-agnostic tool for detecting subspace quality degradation during training. Computing $\gamma$ requires minimal overhead (two norms) and enables early detection of basis drift. For memory-constrained settings (edge devices, federated learning), this diagnostic can guide dynamic dimension adjustment: increase $d$ when $\gamma$ drops, decrease when $\gamma$ stays high.

Beyond memory efficiency, our findings inform understanding of optimization geometry. The success of static bases (despite gradient drift) challenges assumptions underlying adaptive methods and suggests gradient structure may be more stable than previously thought, at least in early-convergence settings.

**Future directions.** Layer-wise subspace methods for deep networks (hierarchical projection structures), theoretical analysis of $\gamma$ evolution dynamics (connecting to loss landscape curvature), extension to transformers and adaptive optimizers (attention-aware subspace design), and large-scale validation on ImageNet and language modeling.

## Reproducibility Statement

All experiments use publicly available CIFAR-10 data. Complete implementation code, configuration files, and analysis scripts are provided in the supplementary material. Training uses standard PyTorch with fixed random seeds for reproducibility. All reported results include standard deviations across 3 independent runs. Raw metrics for all 133 experiments are provided in CSV format.

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

## A. Extended Results: ResNet18

Table 3 presents ResNet18 results. Due to computational constraints, fewer configurations were evaluated.

*Table 3.* ResNet18: Best per Method

| Method | $d$ | Acc | Std |
|---|---|---|---|
| Freq. Directions | 8192 | 67.95 | 0.00 |
| Full SGD | – | 86.83 | 0.75 |

The substantially larger gaps for ResNet18 (18–22 pp vs. 4–6 pp for SmallCNN) suggest that deeper networks may require larger relative subspace dimensions or different tracking strategies.

## B. Alignment Analysis

**Degradation and Gamma Correlation.** Figure 5 shows the relationship between retained variance ($\gamma$) instability and late-epoch performance degradation. Runs with $\gamma_{std} > 0.08$ show degradation exceeding 5%, indicating that consistent subspace quality is important for stable convergence.

**Full Configuration Heatmap.** Figure 6 provides a complete view of accuracy across all evaluated configurations.

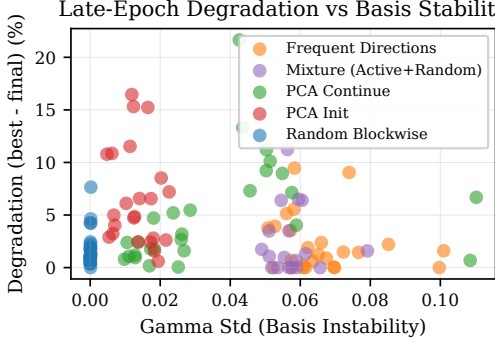

*Figure 5.* Degradation vs. $\gamma$ standard deviation. High instability correlates with larger degradation.



*Figure 6.* Heatmap of test accuracy across methods, dimensions, and learning rates for SmallCNN.

