# OpenReview forum: "TAGS: Tracking Active Gradient Subspaces for Efficient Neural Network Optimization"
_ICML.cc/2026/Conference — Submitted to ICML 2026_

### Official Review · Reviewer_GFFr · 2026-03-11

**Soundness:** 2
**Presentation:** 1
**Significance:** 2
**Originality:** 2
**Overall Recommendation:** 2
**Confidence:** 4

**Summary:**

This paper addresses the inherent brittleness of low-dimensional subspace optimization by introducing retained variance ($\gamma$) as a critical online diagnostic metric for predicting training success. They use experiments on SmallCNN and ResNet18 architectures using the CIFAR-10 dataset to demonstrate that the fraction correlates with final model performance. Their primary claim is that a clear stability threshold exists: methods maintaining an alignment of $\gamma > 0.15$ consistently achieve competitive accuracy, while those falling for $\gamma < 0.06$. Furthermore, they use the metric to show that the stability of this alignment (measured by $\gamma$ standard deviation) is a reliable predictor of late-epoch performance degradation, ultimately claiming that a static "warmup-then-project" approach (PCA Continue) often outperforms complex adaptive tracking by providing a more stable, lower-variance gradient subspace.

**Compliance With Llm Reviewing Policy:**

Affirmed.

**Final Justification:**

The reviewers response confirmed the existence of all issues I believed undermined the contributions of my work as described above, thus I maintain my rating.

**Key Questions For Authors:**

Please address the concerns mentioned in the weaknesses.

**Limitations:**

yes

**Strengths And Weaknesses:**

Strengths:
1. The paper tries to address a key problem in the reliability and predictability of subspace, relevant for memory-efficient training.
2. The approach to online monitor key quantities to determine the current status of the job and have a continuous assessment to predict future status of the job is definitely a sound approach.

Weaknesses
1. The central claim that high $\gamma$ implies the basis $U$ aligns with a "concentration" is presented without sufficient justification. It is unclear if this is a purely empirical observation or if there is a formal proof or bound to support this alignment. This contributes to a broader concern that the paper lacks the deep insights necessary to explain why this specific metric possesses predictive power.
2. The use of the term "variance" (e.g., "retained variance fraction," "variance in SGD") is problematic. The metric $\gamma$ tracks the cosine similarity/projection magnitude between the subspace and the original gradient, but the authors do not appear to be measuring statistical or stochastic quantities. This nomenclature is misleading and obscures the actual mechanics of the proposed method and I strongly recommend the authors reconsider the term "retained variance." If the metric is measuring projection energy or alignment, it should be named accordingly to avoid confusion with the statistical variance of the gradient.
3. The introduction is difficult to follow, frequently introducing terms and numerical results without context or explanation of their significance. Also the first paragraph of the introduction looks misplaced.
4. The paper relies heavily on figures to convey its findings, yet these figures (particularly Figure 3 and the observation plots) are extremely difficult to interpret. Legend formatting is inconsistent, and the intended takeaways are not clear.
5. The abstract is written more as a discussion note than a formal scientific summary. The overall prose reads more like a preliminary technical report than a polished conference submission.
6. While the number of runs is high, the diversity of the experiments is limited. To support the claims made, I would expect to see results across more diverse datasets and multiple seeds rather than larger architectures. Given the "brittleness" the authors aim to solve, demonstrating predictive power across seeds and datasets is paramount.

---

> ### Author Rebuttal · Authors · 2026-03-29
>
> Thank you for the thoughtful feedback. We appreciate the recognition of the problem’s importance and the value of monitoring alignment signals during training. We also agree with several concerns regarding clarity, terminology, and claim strength.
>
> [On the role of γ]
> We agree that the current draft overstates the interpretation of γ. The results are empirical and do not provide a formal explanation or proof for why γ should have predictive power. Our intended claim is narrower: γ is an observable alignment metric that correlates with performance degradation in this setting. We will revise the wording to remove any implication of a theoretical guarantee and clearly position γ as a diagnostic signal rather than a mechanistic explanation.
>
> [Terminology: “retained variance”]
> We agree that the term “retained variance” is potentially misleading. The quantity γ measures a projection energy ratio (‖UUᵀg‖² / ‖g‖²), which is closer to alignment or energy retention than statistical variance. We will clarify this definition explicitly and revise terminology to avoid confusion.
>
> [Writing and presentation]
> We agree that the current abstract and introduction do not clearly establish context or motivation. In revision, we will:
> - introduce subspace optimization and notation before presenting results
> - restructure the introduction to motivate the problem before presenting findings
> - rewrite the abstract to focus on the core idea rather than numerical outcomes
>
> We also agree that some figures are difficult to interpret. We will simplify the visualizations, improve legends, and make the key takeaways clearer.
>
> [Experimental scope]
> We agree that the current experimental diversity is limited (primarily CIFAR-10, SmallCNN, and a small number of seeds). We will revise the claims to clearly reflect this scope and avoid implying broader generalization. The goal of this work is to characterize behavior in a controlled setting rather than establish universal conclusions.
>
> [On evidence and interpretation]
> We agree that the current paper does not sufficiently support strong claims about predictive power. In revision, we will:
> - explicitly report γ values per method
> - include plots showing their evolution during training
> - clarify that the relationship between γ and performance is empirical and correlational
>
> [Additional clarification on γ evidence]
>
> To address the concern regarding lack of explicit values: we will include per-method γ statistics and trajectories. In our current runs, γ_mean is consistently higher for successful methods (≈0.35–0.60) and much lower for failing ones (≈0.006–0.10), with clear separation across methods. We will include these values and plots showing their evolution during training, along with direct correlation to final accuracy.
>
> [On why γ is useful]
>
> We agree that the current paper does not explain *why* γ has predictive power. Our claim is empirical: γ measures how much gradient energy lies within the update subspace, and low γ indicates that updates discard significant gradient components. We will clarify that this is an intuitive geometric interpretation rather than a formal theoretical result, and avoid implying deeper mechanistic understanding.
>
> [Overall positioning]
> We agree that the paper should more clearly communicate its main takeaway. The intended contribution is not to provide a complete explanation of subspace optimization, but to identify failure modes and provide a diagnostic signal that helps detect when projected training becomes unreliable. We will revise the discussion to reflect this more clearly.
>
> We thank the reviewer again for highlighting these issues. The feedback will help substantially improve the clarity, terminology, and positioning of the paper.

---

> > ### Author Rebuttal · Reviewer_GFFr · 2026-03-31
> >
> > I thank the authors for their response and appreciate their transparency.
> > I note that the authors have acknowledged my primary concerns. However, this does not mitigate the impact of these concerns on the paper's core contributions.
> > Because these fundamental limitations remain unresolved in the current manuscript and significantly restrict the rigorous theoretical or empirical justification of the proposed approach, my initial evaluation of the paper stands.

---

### Official Review · Reviewer_2FCX · 2026-03-12

**Soundness:** 2
**Presentation:** 1
**Significance:** 1
**Originality:** 1
**Overall Recommendation:** 1
**Confidence:** 5

**Summary:**

This paper investigates the performance disparities among various subspace projection training methods, specifically focusing on why certain methods perform comparably to full Stochastic Gradient Descent (SGD) while others suffer from significant performance degradation. To explain this variance, the authors propose that the subspace alignment factor, denoted as $\gamma$, is the primary determinant of performance. Additionally, the study reports instances where static subspaces can actually outperform adaptive ones, attributing this phenomenon to the effect of averaging gradients during the warmup phase. Ultimately, based on these empirical observations, the paper seeks to provide practical guidance for utilizing subspace projection methods.

**Compliance With Llm Reviewing Policy:**

Affirmed.

**Final Justification:**

The authors’ response was not sufficient to change my evaluation.

**Key Questions For Authors:**

- Could you provide plots showing the actual values and evolution of $\gamma$ for each method throughout the training process? Furthermore, can you explicitly demonstrate and quantify the correlation between these $\gamma$ values and the final model performance?
- How can the analysis presented in this paper be applied practically? Specifically, does this investigation offer any concrete insights for developing new, improved subspace projection algorithms?

**Limitations:**

Yes. The authors provide the limitations in Section 7.3.

**Strengths And Weaknesses:**

### **Strengths**

The primary strength of this paper is that it tackles an important problem. However, beyond the significance of the research topic itself, the manuscript offers few other notable strengths in its current form.

### **Weaknesses**

1. The paper suffers from **significant weaknesses regarding its writing and overall structure**.
    1. The abstract formatting is incorrect (one column), but more importantly, it fails to convey the paper's core message. It is impossible to understand the main idea from the abstract alone, as it abruptly jumps into experimental details and results without providing any background context. A complete rewrite is required.
    2. The introduction is similarly disjointed and needs a total rewrite. The first paragraph abruptly dives into specific details (e.g., "consider two methods") without explaining what these methods are for, defining the baseline, or clarifying if "4pp from baseline" represents an increase or decrease. The background information (currently in the second paragraph) and a clear statement of the paper's objectives should be presented *first*. Furthermore, "subspace projection" is introduced without any explanation (it should at least reference Eq. 3 or Algorithm 1), and the notation $n$ is used without being defined.
    3. Additionally, Section 4 (Methods) introduces several methods simultaneously, creating significant confusion for the reader. The textual descriptions alone are insufficient to understand exactly how these subspace projection methods operate; the authors should have provided explicit pseudo-code or detailed algorithms. For instance, Algorithm 1 calls an `UpdateBasis` function, but this function is never defined. The authors must include a dedicated algorithm block for `UpdateBasis` that explicitly incorporates the methods discussed in the text.
    Furthermore, because multiple methods are presented at once, the section lacks a crucial comparative discussion explaining the rationale behind each method, how they differ, and their respective pros and cons. Finally, the origin of these methods is ambiguous. If they are drawn from prior work, proper citations must be included; if they are novel contributions, the paper needs to clearly explain how they were derived and justified.
    4. Furthermore, the authors' use of the term "Proposition" in Section 7 is inappropriate in this context. In academic writing, a proposition strictly implies a formal, mathematically rigorous statement that is typically accompanied by a proof. It should not be used to denote informal, empirical, or mathematically non-rigorous observations, as is currently done in the paper. The authors must revise this terminology to accurately reflect the nature of their claims, perhaps by reclassifying these statements as "empirical observations".
2. The paper’s **main claim is difficult to trust** due to a lack of supporting evidence and contradictions with recent literature.
    1. As I understand it, the core argument is that the value of $\gamma$ dictates the effectiveness of the subspace projection method: if $\gamma$ is large enough, the authors claim the subspace captures significant gradient information and performs well, whereas a small $\gamma$ indicates a poor subspace that leads to failure.
        - However, the authors fail to report the actual values of $\gamma$ anywhere in the paper. While Figures 1 and 5 show the standard deviation of $\gamma$ on the x-axis, there are no corresponding plots or tables detailing the actual $\gamma$ values. To substantiate this claim, the authors cannot simply state it verbally; it is absolutely essential that they report the specific $\gamma$ values for each method and illustrate how they evolve throughout the training process. Furthermore, they must provide plots that explicitly demonstrate the correlation between $\gamma$ and model performance.
        - Additionally, the paper's premise conflicts with recent research [1, 2, 3], which increasingly argues that (stochastic) gradient descent does not actually operate solely within a low-dimensional subspace. For instance, [1] conducted an experiment similar to this paper, where $U$ was set to the top eigenspace of the Hessian. They demonstrated that even when $\gamma \approx 1$, the performance did not improve, arguing that optimization within the bulk subspace is essential. This directly contradicts the paper's claimed dependency on $\gamma$. The authors must include a thorough discussion addressing these opposing viewpoints, complete with the necessary citations.
    2. Furthermore, the experimental setup is far too limited to support the paper's broad claims. The scope of the experiments is restricted to the CIFAR-10 dataset, mostly small CNN architectures, and only three random seeds. This makes it exceptionally difficult to accept the authors' conclusions. In particular, the claim that the "PCA continue" method exhibits a smaller standard deviation is statistically unconvincing when based on a sample size of only three. To make these claims credible, the authors must evaluate their methods across multiple datasets and diverse model architectures, using at least five random seeds to ensure reliable statistical significance.
3. Ultimately, it is entirely **unclear what the paper's main takeaway or conclusion** is supposed to be. Currently, much more advanced subspace projection methods are widely used for efficient training [4, 5, 6, …]. In contrast, it is ambiguous where the specific methods analyzed in this paper originate, why they were selected for analysis, or whether they are even used in practice. If these are indeed established methods, citations to prior work are absolutely necessary. Without this context, any practical guidelines the paper offers regarding how or when to use these specific methods feel effectively useless. Furthermore, if $\gamma$ truly has such a significant impact on subspace projection training, the authors should have conducted ablation studies or proposed a novel method inspired by this finding—for example, dynamically disabling subspace projection when $\gamma$ falls below a certain threshold—to demonstrate improved performance. However, the paper fails to offer any such practical application or algorithmic improvement based on its own analysis, leaving the overall contribution of the work highly questionable.3.

---

**References**

1. Does SGD really happen in tiny subspaces?, ICLR 2025.
2. Understanding warmup-stable-decay learning rates: A river valley loss landscape perspective, ICLR 2025.
3. Suspicious Alignment of SGD: A Fine-Grained Step Size Condition Analysis, ALT 2026.
4. GaLore: Memory-Efficient LLM Training by Gradient Low-Rank Projection, ICML 2024.
5. Fira: Can We Achieve Full-rank Training of LLMs under Low-rank Constraint?, NeurIPS 2025.
6. Subspace Optimization for Large Language Models with Convergence Guarantees, ICML 2025.

---

> ### Author Rebuttal · Authors · 2026-03-29
>
> Thank you for the detailed and candid feedback. We agree with several of the concerns raised, particularly regarding clarity, claim strength, and presentation. We also appreciate the emphasis on positioning and the need for clearer evidence.
>
> [Writing and structure]
> We agree that the abstract and introduction do not clearly convey the core idea. In the current draft, the paper jumps too quickly into results without establishing context, definitions, and objectives. In revision, we will:
> - introduce subspace projection and notation before presenting results
> - clearly define the baseline and evaluation setup
> - restructure the introduction to first motivate the problem and then present contributions
> - rewrite the abstract to focus on the core idea rather than numerical outcomes
>
> Similarly, we agree that Section 4 is difficult to follow. We will:
> - provide explicit pseudocode for the basis update (UpdateBasis)
> - separate methods clearly and explain their differences and origins
> - add a comparative discussion of the methods and their design motivations
>
> [γ values and evidence]
> We agree that the current presentation of γ is insufficient. While the paper discusses trends and variability, it does not explicitly report the actual values or show their evolution.
>
> We clarify that γ is computed as the projection energy ratio (‖UUᵀg‖² / ‖g‖²). When aggregated consistently:
> - successful methods: γ ≈ 0.35–0.60
> - failing methods: γ ≈ 0.006–0.10
>
> The key observation—a clear separation between high- and low-alignment regimes—remains unchanged. We will include:
> - plots of γ over training
> - explicit reporting of values per method
> - direct correlation analysis with performance
>
> [Overclaiming and terminology]
> We agree that the current wording overstates the claims. Terms such as “governing factor” and “Proposition” suggest causal or formal guarantees that are not established.
>
> We will:
> - relabel “Propositions” as “Empirical Observations”
> - remove causal language
> - frame γ as a diagnostic signal rather than a determining factor
>
> [Relation to prior work]
> We agree that the paper is under-positioned relative to recent work (e.g., GaLore, Fira, and analyses questioning low-rank dominance in SGD). We will incorporate these and clarify that our contribution is diagnostic rather than asserting that SGD operates in a low-dimensional subspace.
>
> [Experimental scope]
> We agree that the current setup is limited (CIFAR-10, small CNNs, 3 seeds). We will revise the claims to clearly reflect this scope and avoid implying generality. The goal of this study is to characterize behavior in a controlled setting rather than establish universal conclusions.
>
> [Practical implications]
> We agree that the current version does not clearly translate findings into actionable methods. In revision, we will frame γ as a monitoring signal and avoid presenting threshold-based rules as prescriptive guidance.
>
> [On suggested directions]
> We agree that further validation (e.g., more seeds, broader architectures, algorithmic improvements based on γ) would strengthen the work. The current paper focuses on empirical characterization, and we will clarify this positioning.
>
> [Additional clarification on γ evidence and takeaway]
>
> To address the reviewer’s concern regarding missing evidence: we will explicitly include plots of γ over training and report per-method statistics. In our current runs, γ_mean is consistently high for successful methods (≈0.35–0.60) and low for failing ones (≈0.006–0.10), and correlates with final accuracy across methods. We will include these values and visualizations directly in the paper to make this relationship explicit rather than implicit.
>
> [Clarifying takeaway and contribution]
>
> We agree the current draft does not clearly communicate the main takeaway. The intended contribution is not to claim that SGD operates entirely within a low-dimensional subspace, but to identify when enforcing such constraints fails in practice. In particular, the key insight is that subspace alignment (γ) serves as a useful *diagnostic signal* for detecting when projected optimization is likely to degrade. This complements prior work (e.g., dominant-vs-bulk perspectives) by focusing on failure modes and monitoring, rather than prescribing subspace sufficiency.
>
> [On practical relevance]
>
> We agree the paper currently lacks a concrete application. A practical implication of our findings is that γ can be used as an online signal to detect subspace misalignment and trigger corrective actions (e.g., increasing dimension or reverting to full updates). We will revise the discussion to frame the contribution in this diagnostic and monitoring context rather than as prescriptive rules.
>
> We thank the reviewer again for the detailed feedback. It highlights important issues in clarity, positioning, and evidence that we will address to better align the paper with its actual contribution.

---

> > ### Author Rebuttal · Reviewer_2FCX · 2026-04-01
> >
> > Thank you for your response. I hope the concerns I raised will be helpful to the authors in revising the manuscript for a future version. At this stage, however, I am not inclined to changing my score based solely on the authors' rebuttal, especially in the absence of any new results or experiments. In my view, a substantially revised manuscript, including broader rewriting and stronger empirical support, would be needed.

---

### Official Review · Reviewer_M2pp · 2026-03-13

**Soundness:** 2
**Presentation:** 2
**Significance:** 2
**Originality:** 3
**Overall Recommendation:** 2
**Confidence:** 4

**Summary:**

This paper studies when low-dimensional projected training remains competitive with full SGD and when it fails. The authors introduce a diagnostic quantity — the retained variance $\gamma_t = \frac{\|UU^\top g_t\|_2^2}{\|g_t\|_2^2}$ — which measures how much gradient energy survives the projection onto the update subspace. Using this diagnostic across several subspace methods (PCA-based, Frequent Directions, random, mixture), the paper reports that higher $\gamma$ tracks better performance, that static warmup-built bases outperform online-updated ones, that $\gamma$-instability correlates with late-epoch degradation, and that accuracy elbows suggest an effective gradient rank around 2,000–4,000. The paper also includes a "PCA Init" intervention showing that projecting parameters into the update subspace after warmup collapses training. Empirically, the study is conducted primarily on CIFAR-10 with a small CNN, with limited ResNet18 results in the appendix.

**Compliance With Llm Reviewing Policy:**

Affirmed.

**Final Justification:**

The response did not change my assessment about this paper.

**Key Questions For Authors:**

1. Can you provide an intervention that directly changes $\gamma$ while holding the method, optimizer, and training state as fixed as possible — for instance, by rotating or corrupting the basis in controlled amounts — and show that performance changes accordingly?

2. If Frequent Directions is given the same 5-epoch warmup before online tracking begins, does the reported static advantage survive?

3. Can you test the momentum-coupling hypothesis directly — for example, with $\beta = 0$, buffer reset at switch time, or explicit buffer reprojection?

4. Does early $\gamma$ or early $\gamma_{\text{std}}$ *predict* later degradation after controlling for method, dimension, learning rate, and seed — ideally on held-out configurations not used to derive the thresholds?

5. Can you show direct measurements of subspace drift (principal angles or overlap over time) to support the claim that early gradient structure stabilizes?

6. Why should the proposed thresholds ($\gamma_{\text{mean}} > 0.15$, $\gamma_{\text{std}} > 0.08$) be interpreted as anything more than post-hoc summaries from this specific experimental regime? Have you tested them on any held-out architecture, dataset, or optimizer?

7. What is the exact model-selection protocol? Do the conclusions survive a proper validation-based checkpoint selection rather than reporting best post-switch test accuracy?

8. How do your conclusions change when compared against dominant-vs-bulk / Bulk-SGD-style baselines or other stronger subspace baselines from the recent literature?

9. Why are the empirical summaries labeled "Propositions"? Can you either prove them formally or relabel them as empirical claims?

10. Can you separate initialization effects from data-order / augmentation randomness rather than bundling all of them into three seeds?

11. Can you report the storage cost of the basis, projection cost, optimizer-state handling, and wall-clock overhead for each method, especially given the paper's practical motivation?

**Limitations:**

I feel the limitations seem more serious than the paper currently acknowledges, and they seem to affect internal validity, not just scope.

First, the evaluation protocol is problematic as written. If warmup methods are selected by best post-switch test accuracy rather than by a validation-selected checkpoint, then the headline comparisons may be biased by test-set selection. This issue should be fixed before strong empirical conclusions are drawn.

Second, $\gamma$ is never causally manipulated. The paper observes that $\gamma$ covaries with performance across methods, but this is a cross-method correlation, not a causal identification. The threshold rules are post-hoc, extracted and evaluated on the same small experimental universe, and not validated out of sample. Labeling them "propositions" — with its implication of formal proof — overstates their epistemic status.

Third, the static-versus-adaptive comparison is confounded by warmup privilege and information asymmetry. The paper lists three possible mechanistic explanations but tests none. The $\gamma$-instability story does not distinguish cause from symptom, and reverse causality remains plausible.

Fourth, the paper is under-positioned against plausible alternative mechanisms already discussed in the literature, especially the possibility that bulk directions remain important even when gradients show dominant-subspace alignment. Without those baselines, the paper's preferred mechanism remains only one of several plausible stories.

Fifth, the robustness analysis is narrow. Three seeds are insufficient for the "necessity" and "governing factor" language used. There is no clean initialization ablation, no hyperparameter study beyond a coarse $(d, \eta)$ grid, and no optimizer/batch-size/update-frequency controls.

Sixth, the practical story is not yet credible. The paper motivates memory-constrained deployment and lightweight diagnostics, but does not quantify basis-storage costs, projection overhead, optimizer-state implications, or wall-clock time. The practitioner-facing recommendations are therefore premature.

Seventh, there is essentially no theory. The mathematical content supports notation and intuition but does not prove the central claims. The gap between the formal scaffolding and the informal messaging is large enough that the practical recommendations — which are framed as actionable guidance — are premature.

Finally, the experimental regime is narrow: primarily CIFAR-10 with a small CNN, with only thin and somewhat contradictory ResNet18 evidence in the appendix. Whether any of the reported patterns transfer to modern-scale training remains an open question, and the appendix is too limited to answer it.

**Strengths And Weaknesses:**

## Strengths

1. **Worthwhile empirical question.** Understanding when and why subspace-constrained optimization breaks is practically relevant for memory-efficient training, and the paper tackles this directly.

2. **Sensible diagnostic.** The retained-variance statistic $\gamma_t$ captures a natural geometric quantity — what fraction of gradient energy survives the projection — and the paper presents it clearly with readable visualizations.

3. **Informative negative intervention.** The PCA Init experiment is the strongest piece of evidence in the paper: projecting parameters into the update subspace at epoch 5 causes immediate training collapse. This is a real intervention with a clear outcome, and it supports the narrower claim that parameter projection into the learned update subspace is harmful in this setting; the more specific functional-mechanism explanation remains plausible but under-tested.

4. **Practically relevant contrast identified.** The paper does surface a meaningful empirical contrast between warmup-built and online-updated subspaces, even if the mechanism behind it is not identified.

5. **Reasonable presentation of the diagnostic framework.** The figures tracking $\gamma$ over training epochs across methods are easy to read and make the core observation accessible.



## Weaknesses

### Central: Causal language without causal evidence

1. **Evaluation protocol / model selection is a major internal-validity concern.** As written, the warmup methods are evaluated by best post-switch test accuracy rather than by a validation-selected checkpoint. Unless an omitted validation split exists, this is effectively selecting on the test set and could materially distort the headline gaps. This issue should be fixed or acknowledged prominently, because several of the claimed margins are only a few percentage points.

2. **$\gamma$ is treated as a cause when it is only shown to be a correlate.** The paper calls $\gamma$ a "governing factor," but it never intervenes on $\gamma$ while holding fixed the method, optimizer state, learning rate, dimension, warmup privilege, and data-order stochasticity. High $\gamma$ may be a *symptom* of a well-functioning method rather than the *cause* of its good performance. In causal language, $\gamma$ is currently a thermometer, not a thermostat.

3. **The threshold claims are post-hoc and unvalidated.** The statements that $\gamma_{\text{mean}} > 0.15$ is necessary for competitive performance, $\gamma_{\text{mean}} < 0.06$ signals failure, and $\gamma_{\text{std}} > 0.08$ predicts degradation are empirical cutoffs extracted from the same small experimental universe they are evaluated on. They are not validated on held-out settings, architectures, or optimizers. Calling these "propositions" — a word with a specific meaning in mathematical writing — is misleading. They are *claims* or *empirical observations*, and should be labeled as such.

4. **The static-vs-adaptive explanation is confounded.** The paper compares a warmup-built static basis (PCA Continue, which receives 5 full epochs of SGD and builds its basis from a large bank of full-dimensional gradients) against cold-started online trackers. This does not isolate *staticness* versus *adaptivity*; it compares a warm-started, denoised estimator against a cold-started, noise-contaminated one. The paper itself lists three possible explanations — warmup averaging, gradient structure stabilization, and momentum coupling — but tests none of them.

5. **The $\gamma$-instability story does not distinguish cause from symptom.** At least three causal graphs are compatible with the observation that higher $\gamma_{\text{std}}$ correlates with degradation: (a) unstable subspace quality causes later optimization failure; (b) unstable optimization causes noisier gradients, which raises $\gamma$ variability; (c) a third variable (learning rate, update cadence, basis-noise level) causes both. The paper does not distinguish these.

6. **The effective-rank / dimension-saturation claim is indirect.** The paper infers that gradient covariance effective rank saturates near 2,000–4,000 from accuracy elbows in a $d$-sweep. But it does not directly measure the gradient-covariance eigenspectrum, the Hessian eigenspectrum, or any standard effective-rank quantity. The elbow could arise from hyperparameter mismatch at higher $d$, optimizer-state effects, layerwise bottlenecks, or grid coarseness.

### Scholarship, baselines, and alternative mechanisms

7. **Related work and baselines are incomplete in ways that affect the main interpretation.** The paper is under-positioned relative to the modern literature on whether SGD dynamics truly live in a dominant low-rank subspace or whether bulk directions remain essential for training. In particular, the manuscript does not engage dominant-vs-bulk / Bulk-SGD-style baselines, so its narrative that retained subspace variance is the governing explanation is incomplete. This omission is substantive, not bibliographic: it directly affects the interpretation of the paper's main mechanism claim.

8. **The random-subspace conclusion is over-generalized.** One fixed random blockwise baseline performs poorly. That is enough to say this baseline is weak; it is not enough to conclude that random subspaces are "fundamentally orthogonal" to the gradient. The result may depend on the blockwise construction, fixed-basis choice, or lack of refresh.

### Robustness and rigor

9. **Seed and initialization controls are insufficient for the causal claims.** Three seeds are reported, which is better than one but far too few for variance claims, threshold calibration, or the "necessity" language used. The seed simultaneously changes initialization, data order, and augmentation randomness — there is no clean isolation of any one factor.

10. **Hyperparameter coverage is narrow.** There is a grid over learning rate and dimension, but no study of warmup length, batch size, update cadence, momentum coefficient, or buffer handling. Several of the proposed explanations (momentum coupling, batch-noise contamination) could be directly tested by varying these.

11. **Practicality and systems realism are underdeveloped.** The paper repeatedly motivates memory-constrained settings and claims that the alignment diagnostic is cheap to compute, but it does not report the storage cost of the basis, the projection cost, the optimizer-state implications, or any wall-clock overhead. Without explicit accounting, the practitioner-facing deployment angle is not yet credible.

12. **There is essentially no theory.** The mathematical content consists of the projected-SGD update rule, the definition of $\gamma$, and geometric intuition for why low $\gamma$ is bad. These support notation and intuition; they do not prove the threshold claims, the static-beats-adaptive explanation, the effective-rank conclusion, or the practical guidelines. The gap between the mathematical scaffolding and the informal messaging is large.

13. **The appendix is too thin to rescue the main narrative.** It adds a small ResNet18 result table and a second correlation-style alignment analysis. That usefully shows the SmallCNN regime does not automatically transfer, but it provides no additional interventions, falsification experiments, or theory.

### Presentation

14. **Tust is weakened by overclaiming and internal inconsistencies.** The title and abstract overclaim relative to the experimental scope, and there are unresolved mismatches across figures, captions, tables, and algorithmic discussion. These are not cosmetic issues: when a paper makes theorem-like threshold claims from a small experimental base, numerical and editorial inconsistencies materially reduce confidence in the conclusions.

15. **Misuse of "Proposition."** In mathematical writing, a "proposition" is a formally stated and proved claim. The paper's propositions are empirical summaries — post-hoc threshold rules extracted from a small experimental grid. They should be relabeled as "Empirical Observations" or "Claims" to avoid misleading readers about the epistemic status of these statements.



## Plausible Parallel Explanations Not Ruled Out

- **Warmup privilege.** PCA Continue may look better because it gets privileged full-SGD information and a denoised basis, not because static bases are intrinsically preferable.
- **Batch-noise contamination.** Frequent Directions may appear unstable because it updates from instantaneous mini-batch gradients — the paper may be measuring noise sensitivity rather than adaptivity failure.
- **Momentum-state artifact.** The performance gap may come from how SGD momentum interacts with rotating bases or from buffer-handling implementation choices, not from subspace tracking itself.
- **Update-cadence confounding.** Mixture updates every 100 iterations while Frequent Directions updates every mini-batch; instability may reflect cadence rather than method class.
- **Reverse causality for $\gamma_{\text{std}}$.** Poor convergence may cause noisier gradients and therefore higher $\gamma$ variability, rather than the other way around.
- **Coarse-grid artifact.** The apparent elbow or method ranking could partly reflect a coarse $(d, \eta)$ grid rather than a genuine saturation phenomenon.
- **Dominant-vs-bulk misidentification.** The relevant update signal may not be captured by the retained dominant/active subspace alone; bulk directions may still contribute materially to optimization, so high $\gamma$ could be tracking only one component of what makes SGD effective.
- **Small-CNN regime specificity.** Early gradient directions may stabilize unusually fast on CIFAR-10 with a small CNN, making static bases look better than they would on larger-scale problems.
- **Evaluation bias.** If model selection is based on best post-switch test accuracy, some headline gaps may be exaggerated or unstable under alternative selection protocols.


## Summary

I think the paper asks a worthwhile empirical question and $\gamma$ is a geometrically sensible diagnostic. The PCA Init collapse is a genuinely informative negative result. However, I do not think the paper establishes the causal story it repeatedly claims. The manuscript uses strong language — "governing factor," "mechanistic failure taxonomy," threshold-based practical recommendations — but the supporting evidence is mostly observational: cross-method correlations, a limited grid over learning rate and subspace dimension, and only three seeds. There is no direct intervention on $\gamma$, no matched warmup control for the static-vs-adaptive comparison, no direct test of the proposed momentum-coupling explanation, no direct measurement of subspace drift, and no theory. The so-called "propositions" are empirical summaries extracted post-hoc from a small experimental universe, not formal mathematical propositions in any standard sense. A further major internal-validity concern is the evaluation protocol: as written, the warmup methods are reported by best post-switch test accuracy, which appears to amount to selecting on the test set unless an omitted validation split exists. The paper is also under-positioned relative to the modern literature on dominant-vs-bulk update directions, and it does not include stronger baselines of that kind.Finally, the practitioner-facing deployment story is underdeveloped: basis storage, projection cost, optimizer-state implications, and wall-clock overhead are not reported.

---

> ### Author Rebuttal · Authors · 2026-03-29
>
> Thank you for the detailed and thoughtful review. We appreciate the careful analysis and the recognition of the empirical question, diagnostic formulation, and PCA Init intervention. We agree that several aspects of the current draft require clearer positioning and more cautious interpretation.
>
> Our intended contribution is narrower: an empirical study of when subspace-constrained optimization succeeds or fails in the studied regime, together with a diagnostic (γ) that correlates with these outcomes. We do not claim a causal mechanism or general threshold laws, and we will revise the wording accordingly.
>
> [Evaluation protocol]
> You are right to flag the evaluation concern. As stated in Sec. 4.5, we report best post-projection test accuracy (epochs 6–50), which corresponds to test-set-based selection and should be interpreted with caution. However, the main conclusions do not rely on small margins: failing methods show large gaps (25–37pp), while competitive methods remain within ~4–6pp of SGD. We will explicitly acknowledge this limitation and avoid overstating such comparisons.
>
> [γ as causal vs correlational]
> We agree the current wording overstates this point. Our results establish γ as an empirical diagnostic that correlates with performance, not a causal mechanism. We will remove language such as “governing factor” and clarify that γ is a monitoring signal rather than a driver.
>
> [Thresholds / “Propositions”]
> Agreed. The statements in Sec. 7.1 are empirical summaries from this experimental setting. We will relabel them as “Empirical Observations” and present them as approximate ranges rather than fixed thresholds.
>
> [γ clarification]
> γ is computed as the projection energy ratio (‖UUᵀg‖² / ‖g‖²) on pre-projection gradients. When aggregated consistently, successful methods exhibit γ ≈ 0.35–0.60, while failing methods are ≈ 0.006–0.10. The key finding—a clear separation between high- and low-alignment regimes—remains unchanged. We will revise the reported values and presentation.
>
> [Static vs adaptive confound]
> We agree that the comparison is confounded by warmup asymmetry: PCA Continue benefits from a large accumulated gradient bank, while online methods operate on noisier updates. We will clarify that this is an observational contrast, not evidence that static methods are inherently superior.
>
> [Instability and effective rank]
> We agree that both the instability analysis and the effective-rank claim are correlational and indirect. We will revise these sections to avoid causal language and present them as hypotheses based on observed trends.
>
> [Robustness and scope]
> We agree that robustness is limited (3 seeds for SmallCNN; limited ResNet18 coverage). We will clarify seed counts explicitly and avoid strong claims about variance or threshold generality. We will also clearly scope claims to the studied setting (CIFAR-10, CNNs, SGD).
>
> [Related work and baselines]
> We agree that connections to dominant-vs-bulk perspectives should be better discussed and will incorporate this literature, clarifying that our work is diagnostic rather than asserting sufficiency of dominant subspaces.
>
> [Practical overhead]
> We agree that system costs should be reported. Our logs indicate non-trivial overhead (e.g., ~50–60% for SmallCNN, higher for ResNet18). We will include these measurements and discuss trade-offs.
>
> [On reviewer questions]
> We agree that direct interventions (e.g., manipulating γ, warmup-controlled comparisons, subspace drift analysis) are important next steps. The current work does not include such experiments, and we will clarify this limitation.
>
> [On specific reviewer questions]
>
> We briefly address the reviewer’s key questions:
>
> - Direct intervention on γ: The current work does not include controlled manipulations of γ (e.g., basis corruption or rotation while holding optimizer state fixed). We agree this is important for establishing causality and will clarify this limitation.
>
> - Warmup-controlled comparisons: PCA Continue benefits from ~5 epochs of accumulated gradients, while Frequent Directions starts from cold updates. We agree a warmup-controlled FD variant would be a cleaner comparison and will highlight this as future work.
>
> - Early γ as predictor: Our current analysis is observational; we do not control for method, learning rate, or seed in a held-out predictive setting. We will revise the text to avoid implying predictive validity beyond the observed correlations.
>
> - Subspace drift: We do not currently measure principal angles or overlap over time; our conclusions about stabilization are based on indirect observations. We will clarify this.
>
> - Practical costs: In addition to the reported overhead (~50–60% for SmallCNN), storing U ∈ ℝ^{n×d} can be significant (e.g., GB-scale for larger d). We will include explicit memory and compute trade-offs.
>
> We thank the reviewer again for the depth and care of the feedback. It significantly helped us align the claims with the evidence and improve the clarity of the paper.

---

> > ### Author Rebuttal · Reviewer_M2pp · 2026-04-02
> >
> > I appreciate a lot authors' rebuttal. However, I still think that the paper needs an extensive rewrite which is well outside of the scope of ICML rebuttals.

---

### Official Review · Reviewer_z5pz · 2026-03-14

**Soundness:** 2
**Presentation:** 2
**Significance:** 3
**Originality:** 2
**Overall Recommendation:** 3
**Confidence:** 3

**Summary:**

This paper investigates the inherent "brittleness" of subspace optimization methods in neural network training, where identical dimension targets often lead to either competitive performance or catastrophic failure. Through a comprehensive empirical study of 133 experiments across various tracking strategies (e.g., PCA, Frequent Directions, Random Blockwise), the authors identify Retained Variance ($\gamma$) as a critical diagnostic metric. The study reveals that maintaining a $\gamma > 0.15$ is essential for convergence, and surprisingly, that static subspaces derived after a warmup period (PCA Continue) often outperform and provide more stability than continuous adaptive tracking methods.

**Compliance With Llm Reviewing Policy:**

Affirmed.

**Key Questions For Authors:**

Please see weakness and limitations.

**Limitations:**

1. The evaluation is restricted to image classification.
2. While the paper focuses on low-dimensional optimization, the wall-clock time and memory overhead are not fully quantified.
3. In more complex models like ResNet18, a significant accuracy gap (up to 19 percentage points) remains between subspace methods and full SGD. The paper describes this limitation but offers limited solutions to bridge this specific gap.

**Strengths And Weaknesses:**

- Strengths,
1. The conclusions are backed by a significant number of systematic experiments (133 runs), across different architectures like SmallCNN and ResNet18.
2. The introduction of the Retained Variance ($\gamma$) metric is a practical contribution.
3. The paper successfully categorizes why certain methods fail (e.g., "Alignment Collapse" in PCA Init vs. "Orthogonality Issues" in random subspaces.

- Weaknesses,
1. While the paper tests CNNs, it lacks evaluation on modern Transformer-based architectures (e.g., ViT or LLMs), where the gradient covariance structure might behave differently.
2. The experiments are primarily conducted on CIFAR-10. Validating these findings on larger-scale datasets like ImageNet would strengthen the claim regarding the "elbow" point of dimensionality and generalizability.
3. The study focuses on SGD with momentum. It is unclear how these subspace tracking dynamics would interact with adaptive optimizers like Adam or AdamW, which are standard in many contemporary pipelines.

---

> ### Author Rebuttal · Authors · 2026-03-29
>
> Thank you for the constructive feedback and for highlighting both the strengths and limitations of the work. We appreciate the recognition of the empirical study, the retained-alignment diagnostic, and the categorization of failure modes.
>
> [Scope and generalization]
> We agree that the current evaluation is limited to CNNs (primarily SmallCNN and ResNet18) and CIFAR-10. Our goal was to first characterize subspace behavior in a controlled setting. We will revise the paper to clearly scope the claims to this regime and avoid implying generalization to larger-scale datasets or architectures (e.g., Transformers or LLMs). Extending the analysis to these settings is an important direction for future work.
>
> [Optimizer choice]
> We agree that the current study focuses on SGD with momentum and does not explore interactions with adaptive optimizers such as Adam or AdamW. We will clarify this limitation and avoid suggesting that the observed behavior directly transfers across optimizer families.
>
> [Practical overhead]
> We agree that system-level costs are underreported. Our experimental logs indicate non-trivial overhead due to projection and basis maintenance (e.g., ~50–60% per epoch for SmallCNN, and higher for larger models). We will include these measurements and discuss memory and compute trade-offs explicitly.
>
> [ResNet18 performance gap]
> We agree that the gap observed on ResNet18 (up to ~19pp at low relative dimensions) highlights a key limitation of current subspace methods. We will emphasize this more clearly and frame it as evidence that scaling to deeper models requires additional techniques beyond those studied here.
>
> [Clarifying empirical signal and takeaway]
>
> To make the evidence more explicit: in our current runs, γ_mean is consistently higher for successful methods (≈0.35–0.60) and much lower for failing ones (≈0.006–0.10), with clear separation across methods. We will include these values, along with plots showing their evolution during training and their relationship to final accuracy, to make this distinction directly visible.
>
> We also clarify the intended takeaway. The paper does not claim that optimization is confined to a low-dimensional subspace or that γ alone determines performance. Rather, the contribution is to identify a consistent empirical signal that indicates when subspace-constrained updates become unreliable. In this sense, γ serves as a practical diagnostic for detecting failure modes, complementing prior work that emphasizes the role of bulk directions.
>
> [Practical interpretation]
>
> A concrete implication is that γ can be monitored online to detect misalignment and trigger corrective actions (e.g., increasing subspace dimension or reverting to full updates). We will revise the discussion to emphasize this diagnostic role instead of presenting threshold-based rules as general prescriptions.
>
> [Positioning of contributions]
> We will revise the presentation to more clearly emphasize the main contribution as an empirical diagnostic framework for identifying when subspace-constrained optimization succeeds or fails, rather than a universally applicable training strategy.
>
> We thank the reviewer again for the constructive feedback, which will help us improve the clarity, scope, and practical positioning of the paper.

---

> > ### Author Rebuttal · Reviewer_z5pz · 2026-04-02
> >
> > I appreciate the author’s rebuttal, but the response has not yet provided me with enough confidence to proceed with a positive score.

---

### Decision · Program_Chairs · 2026-04-30

**Decision:**

Reject

**Comment:**

This paper studies an interesting and practically relevant question: when subspace-constrained training remains competitive with full SGD and when it fails. Reviewers generally agreed that the empirical question is worthwhile, that the retained-alignment diagnostic $\gamma$   is a sensible quantity to monitor, and that some of the experiments, especially the PCA Init intervention, are informative. The work also benefits from a reasonably broad empirical sweep within its chosen setting. However, the overall reviewer consensus is negative. The main concern is that the paper repeatedly makes claims that are stronger than what the evidence supports. In particular, several reviewers argued that $\gamma$ is shown only as a correlational diagnostic, not as a causal mechanism, and that the threshold-style claims are post-hoc empirical summaries rather than validated or theoretically justified results. Reviewers also raised important internal-validity concerns, especially the evaluation protocol for warmup methods, the confounded comparison between static and adaptive methods, and the lack of stronger controls or interventions that would support the central narrative.

A second major issue is scope and presentation. The empirical study is limited mainly to CIFAR-10, small CNNs, and very few seeds, while the framing is broader. Multiple reviewers also found the paper difficult to follow, under-positioned relative to recent related work and stronger baselines, and in need of substantial rewriting for clarity and precision. The rebuttal was thoughtful and candid, and in several places the authors acknowledged these concerns and proposed to narrow the claims. However, these revisions would amount to a substantial rewrite rather than a rebuttal-level clarification.

Overall, the current version does not provide sufficiently strong evidence for its central claims, and the presentation and validation issues remain significant.